# Quantum noise and its evasion in feedback oscillators

Hudson A. Loughlin [1] ✉ & Vivishek Sudhir[1,2] ✉

Feedback oscillators, consisting of an amplifier whose output is partially fed back to its input, provide stable references for standardization and synchronization. Notably, the laser is such an oscillator whose performance can be limited by quantum fluctuations. The resulting frequency instability, quantified by the Schawlow-Townes formula, sets a limit to laser linewidth. Here, we show that the Schawlow-Townes formula applies universally to feedback oscillators beyond lasers. This is because it arises from quantum noise added by the amplifier and out-coupler in the feedback loop. Tracing the precise origin of quantum noise in an oscillator informs techniques to systematically evade it: we show how squeezing and entanglement can enable sub-Schawlow-Townes linewidth feedback oscillators. Our analysis clarifies the quantum limits to the stability of feedback oscillators in general, derives a standard quantum limit (SQL) for all such devices, and quantifies the efficacy of quantum strategies in realizing sub-SQL oscillators.

Feedback oscillators[1–3] suffuse the modern world, and their stability[4,5] demarcates what is possible in every conceivable enterprise. Frequency fluctuations of oscillators limit measurements of distance—using a radar[6], lidar[7], or for gravitational-wave detection[8–10]—and of time using optical atomic clocks[11]. The performance of information processors—classical[12–14] or quantum[15,16]—and communication systems[17–19] are also limited by the stability of their clocks. So is our sensitivity to new physics—at low energy[20] or using high-energy colliders[21]; and of our planet[22,23], or the universe[24,25]. Even modern economic practice is beholden to the ticks of a clock[26].

A paradigmatic example of a feedback oscillator whose stability is well understood is the laser[27,28]. Its frequency noise is given by the (modified) Schawlow-Townes (ST) formula[29–33],

$$\bar{S}_{\dot{\varphi}\dot{\varphi}}^{ST}[\Omega] = \frac{\hbar\Omega_0\kappa^2}{P_0}\left(\frac{1}{2} + n_{th}\right) = \left(\frac{\ln\eta}{\tau|\alpha|}\right)^2\left(\frac{1}{2} + n_{th}\right), \tag{1}$$

for the symmetrized double-sided spectrum of the frequency deviation $\dot{\varphi}$ from the oscillator's nominal output frequency $\Omega_0$. Here, $P_0 = \hbar\Omega_0|\alpha|^2$ is the oscillator's mean output power and $\alpha$ is the amplitude of the mean photon flux. In a laser, the feedback element is a cavity, whose round-trip time is $\tau$, average thermal occupation is

$n_{th} = \left[\exp(\hbar\Omega_0/k_BT) - 1\right]^{-1}$, and light is out-coupled through a mirror with power reflectivity $\eta$, equivalent to the cavity linewidth $\kappa = \tau^{-1}\ln\eta$. The Schawlow-Townes formula is more commonly specified as a linewidth instead of a spectral density. The full-width-at-half-maximum linewidth $\Gamma$ of a signal with flat frequency noise spectrum is[34], $\Gamma = \bar{S}_{\dot{\varphi}\dot{\varphi}}/(2\pi)$; so the linewidth corresponding to eq. (1) is

$$\Gamma_{ST} = \frac{1}{2\pi}\left(\frac{\ln\eta}{\tau|\alpha|}\right)^2\left(\frac{1}{2} + n_{th}\right). \tag{2}$$

We henceforth confine attention to the performance of quantum-noise-limited oscillators, corresponding to the case $n_{th} = 0$ (we also neglect corrections known to arise from the coupling of temporal and/or spatial modes[35]).

In this work, we show that the Schawlow-Townes limit applies to a much larger class of feedback oscillators: those that are constructed by positive feedback of the output of a broadband amplifier. In doing so, we precisely identify the origin of the ST limit, extend the well-known theory of quantum noise in amplifiers[36,37] to oscillators, and establish a bridge between the classical electronic theory of feedback oscillators and the quantum electronics of a laser. Based on these insights, we

[1]LIGO Laboratory, Massachusetts Institute of Technology, Cambridge, MA 02139, USA. [2]Department of Mechanical Engineering, Massachusetts Institute of Technology, Cambridge, MA 02139, USA. ✉e-mail: hudsonl@mit.edu; vivishek@mit.edu

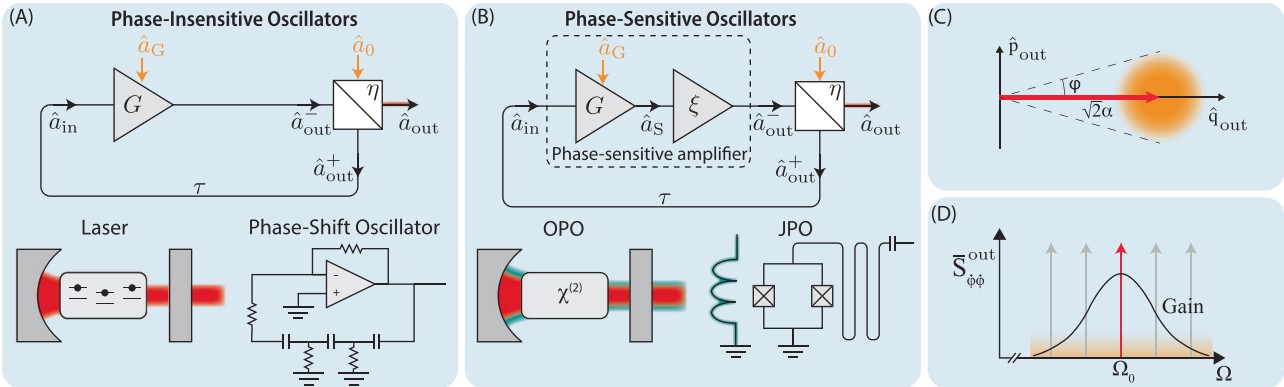

**Fig. 1 | Physical and abstracted feedback oscillators. A** shows a minimal abstract representation of oscillators based on phase-insensitive amplifiers, where the amplifier output is fed back with a unitary element, here modeled by a beam-splitter. Physical examples of such oscillators include lasers and phase-shift oscillators. **B** shows a similar abstraction for the phase-sensitive case where the output of a phase-sensitive amplifier (dashed rectangle, composed of a phase-insensitive amplifier followed by an ideal squeezer) is fed back. Physical examples of phase-sensitive oscillators include optical parametric oscillators (OPOs) and Josephson parametric oscillators (JPOs). **C** is a phase-space diagram showing a sketch of the carrier field (red) and vacuum fluctuations (orange) in the two quadratures $\hat{q}, \hat{p}$. These fluctuations cause the oscillators' amplitude and phase to fluctuate. **D** shows a frequency-domain picture of the carrier (red) that is selectively amplified by the overlapping gain profiles of the in-loop amplifier (black) and the resonant harmonics of the feedback loop (gray).

then consider methods of evading the ST bound by manipulating the oscillators' quantum states. We find that squeezing, entanglement, and phase-sensitive amplification in feedback oscillators allow us to achieve stability beyond the Schawlow-Townes limit, providing a new avenue toward realizing ultra-stable oscillators.

## Results

### Quantum feedback oscillators
We consider, in Fig. 1A, the simplest configuration of a feedback oscillator consistent with the laws of quantum physics: a phase-insensitive amplifier with linear gain $G$ embedded in a feedback loop of time delay $\tau$, whose output, because of the no-cloning theorem[38], is extracted from the loop using a beam-splitter. Clearly, the mode $\hat{a}_0$ adds quantum noise at the out-coupler.

In the absence of the feedback loop, the observation of Haus-Caves[36,37] is that the classical input-output relation $\langle\hat{a}_{\text{out}}\rangle = G\langle\hat{a}_{\text{in}}\rangle$, cannot be promoted to the relation between operators $\hat{a}_{\text{out}}(t) = G\hat{a}_{\text{in}}(t)$, since that is inconsistent with the commutation relations (here $i,j \in \{\text{in, out}\}$)

$$[\hat{a}_i(t), \hat{a}_j^\dagger(t')] = \delta(t-t')\delta_{ij}. \tag{3}$$

Consistency can be achieved by modifying the input-output relation to

$$\hat{a}_{\text{out}}^-(t) = G\hat{a}_{\text{in}}(t) + \sqrt{G^2-1}\,\hat{a}_G^\dagger(t), \tag{4}$$

where the ancillary mode $\hat{a}_G$ is such that $[\hat{a}_G(t), \hat{a}_G^\dagger(t')] = \delta(t-t')$ and $\langle\hat{a}_G(t)\rangle = 0$. Physically, it conveys unavoidable noise added by the amplifier's internal degrees of freedom[32,39].

These observations do not apply when the feedback loop is closed. First, positive feedback will lead to a large mean field at the input of the amplifier, which will saturate its output due to the intrinsic nonlinearity of the amplifier. Note that nonlinearity in the response is a fundamental requirement of any physical amplifier since its gain arises from an external source whose energy density has to be finite; it is precisely this nonlinearity that will determine the amplitude of the oscillating output field (as discussed below). Second, the commutation relations in eq. (3) only apply to freely propagating fields, not those inside a feedback loop[40,41]. We will now resolve these issues and show that a proper account of the saturation behavior leads to the threshold condition for oscillation: "gain = loss", while a proper imposition of the commutation relation gives the correct quantum noise of the oscillator.

### Saturation, steady-state, and linear gain
The nonlinear input-output behavior of the (memoryless) amplifier can be cast as

$$\alpha_{\text{out}}^-(t) = \mathcal{A}(\alpha_{\text{in}}(t)), \tag{5}$$

where, $\alpha_i = \langle\hat{a}_i\rangle$ ($i \in \{\text{in, out}\}$) is the mean amplitude, and $\mathcal{A}(\cdot)$ is the nonlinear gain function which we postulate has the following properties: (1) $\mathcal{A}(-x) = -\mathcal{A}(x)$, i.e., the amplifier is symmetric and bipolar; (2) $\frac{d\mathcal{A}}{dx} > 0 \quad \forall x$, i.e., the amplifier's output is a monotonically increasing function of its input; (3) $\mathcal{A}(x \to 0) \to G_0 x$ for some $G_0 > 1/\sqrt{\eta}$, i.e., there exists a "small signal" regime of linear gain $G_0$, such that this gain is larger than the loss via the out-coupler, parametrized by $\sqrt{\eta}$; and, (4) $\frac{d^2\mathcal{A}}{dx^2} < 0$, i.e., the instantaneous gain, $\frac{d\mathcal{A}}{dx}$, is a monotonically decreasing function of the amplifier's input. (The weaker condition that $\frac{d^2\mathcal{A}}{dx^2} < 0$ at the smallest value of $x$ for which $\frac{d\mathcal{A}}{dx} = 1/\sqrt{\eta}$ is sufficient to ensure stability against infinitesimal perturbations; condition (4) ensures stability against finite perturbations, see "Methods" subsection "Saturating behavior of phase-insensitive feedback oscillators and classical steady state").

The output of the amplifier $\alpha_{\text{out}}^-(t)$ propagates through the feedback loop and appears as

$$\alpha_{\text{in}}(t) = -\sqrt{\eta}\,\alpha_{\text{out}}^-(t-\tau) + \sqrt{1-\eta}\,\alpha_0(t-\tau) \tag{6}$$

at the input of the amplifier. The classical, nonlinear, input-output relations, eqs. (5) and (6), together with the postulates of the nonlinear gain, $\mathcal{A}$, suffice to determine the oscillator's classical steady-state. As discussed in detail in "Methods", the oscillator will begin at an unstable equilibrium point with zero output field amplitude. Any perturbation, such as noise, will then kick the oscillator away from the unstable equilibrium and initiate oscillations, just as in a laser[42]. Small signal analysis shows that this oscillatory state is stable if the in-loop field amplitude, $\alpha_{\text{ss}}$, satisfies

$$|\mathcal{A}[\alpha_{\text{ss}}]| = \alpha_{\text{ss}}/\sqrt{\eta}. \tag{7}$$

The corresponding output, $\alpha = \sqrt{\eta}\alpha_{\text{ss}}$, determines the oscillator's amplitude and serves as the phase reference for quantum fluctuations

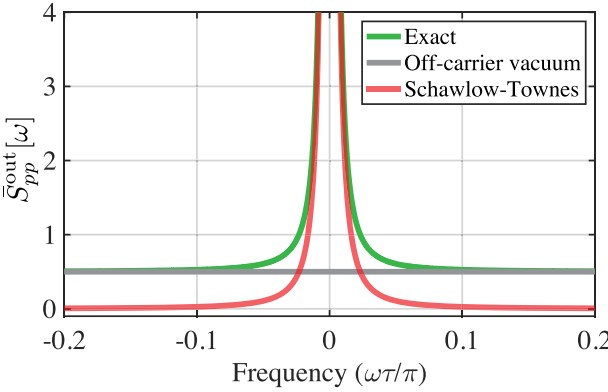

**Fig. 2 | Phase noise of feedback oscillator.** The output phase quadrature spectrum of a feedback oscillator with a phase-insensitive in-loop amplifier is well approximated near the carrier by a second-order pole, producing a Lorentzian line shape. Red shows the Schawlow-Townes component of the line shape near the carrier, while green shows the full prediction [eq. (16)] including the white vacuum noise far from carrier (gray line).

which will be discussed throughout this paper. The linear gain of the amplifier in this steady state is (see "Methods" subsection "Saturating behavior of phase-insensitive feedback oscillators and classical steady state")

$$G \equiv \lim_{t \to +\infty} \alpha_{\text{out}}^{-}(t)/\alpha_{\text{in}}(t) = 1/\sqrt{\eta}. \tag{8}$$

This equation has the form "gain = loss" and defines the linear gain seen by the quantum fluctuations that ride on top of the classical steady-state.

**Quantum fluctuations**

Around the steady state, the (Heisenberg-picture) operators representing the quantum fluctuations satisfy the set of linear equations:

$$
\begin{aligned}
\hat{a}_{\text{out}}^{-}[\Omega] &= G\,\hat{a}_{\text{in}}[\Omega] + \sqrt{G^2-1}\,\hat{a}_{\text{G}}^{\dagger}[\Omega] \\
\hat{a}_{\text{out}}^{+}[\Omega] &= -\sqrt{\eta}\,\hat{a}_{\text{out}}^{-}[\Omega] + \sqrt{1-\eta}\,\hat{a}_0[\Omega] \\
\hat{a}_{\text{out}}[\Omega] &= \sqrt{1-\eta}\,\hat{a}_{\text{out}}^{-}[\Omega] + \sqrt{\eta}\,\hat{a}_0[\Omega] \\
\hat{a}_{\text{in}}[\Omega] &= e^{i\Omega\tau}\,\hat{a}_{\text{out}}^{+}[\Omega],
\end{aligned}
\tag{9}
$$

which we express in terms of their Fourier transforms. Here $G$ is the linear gain of the amplifier, the ancillary mode $\hat{a}_{\text{G}}$ describes the unavoidable noise associated with amplification, and the last equation is the Fourier transform of the time-domain relation, $\hat{a}_{\text{in}}^{-}(t) = \hat{a}_{\text{out}}^{+}(t-\tau)$, expressing the delay in the feedback path. Note that we take the amplifier's gain to be frequency-independent for frequencies comparable to the inverse delay of the loop.

The quantum statistics of the ancillary mode, $\hat{a}_{\text{G}}$, in closed-loop operation (i.e., $0 < \eta < 1$) need not coincide with those in open-loop operation ($\eta = 0$)[36,37,43]. In the closed-loop configuration, since the ancillary mode is not freely propagating, it need not satisfy the commutation relations (the Fourier transform analog of eq. (3))

$$[\hat{a}_i[\Omega], \hat{a}_j^{\dagger}[\Omega']] = 2\pi \cdot \delta[\Omega + \Omega']\delta_{ij}. \tag{10}$$

of a freely propagating field[40,41]. However, the fields in-coupled and out-coupled from the oscillator, $\hat{a}_0$ and $\hat{a}_{\text{out}}$, are freely propagating and must obey eq. (10). To enforce this constraint, we solve eq. (9) expressing the output in terms of the in-coupled and ancillary fields:

$$\hat{a}_{\text{out}}[\Omega] = H_0[\Omega]\hat{a}_0[\Omega] + H_{\text{G}}[\Omega]\hat{a}_{\text{G}}^{\dagger}[\Omega], \tag{11}$$

where the transfer functions are

$$H_0[\Omega] = \frac{\sqrt{\eta} + e^{i\Omega\tau}/\sqrt{\eta}}{1 + e^{i\Omega\tau}}, \quad H_{\text{G}}[\Omega] = \frac{1/\sqrt{\eta} - \sqrt{\eta}}{1 + e^{i\Omega\tau}}. \tag{12}$$

Here we have assumed steady-state operation in which [eq. (8)] $G\sqrt{\eta} = 1$. Insisting that $\hat{a}_{\text{out}}$ and $\hat{a}_0$ satisfy eq. (10), forces the ancillary mode to satisfy

$$
\begin{aligned}
[\hat{a}_{\text{G}}[\Omega], \hat{a}_{\text{G}}^{\dagger}[\Omega']] &= \frac{|H_0[\Omega]|^2 - 1}{|H_{\text{G}}[\Omega]|^2} 2\pi \cdot \delta[\Omega + \Omega'] \\
&= 2\pi \cdot \delta[\Omega + \Omega'],
\end{aligned}
\tag{13}
$$

where we have used the fact that $|H_0[\Omega]|^2 - |H_{\text{G}}[\Omega]|^2 = 1$. Thus, although $\hat{a}_{\text{G}}$ is not freely propagating, it obeys the usual canonical commutation relation.

**Output spectrum and Schawlow-Townes formula**

Our interest is in the amplitude and phase quadratures of the output field. For an arbitrary field $\hat{a}_i$, these are defined by $\hat{q}_i = (\hat{a}_i^{\dagger} + \hat{a}_i)/\sqrt{2}$ and $\hat{p}_i = i(\hat{a}_i^{\dagger} - \hat{a}_i)/\sqrt{2}$. Equation (11) can then be written as

$$
\begin{aligned}
\hat{q}_{\text{out}}[\Omega] &= H_0[\Omega]\hat{q}_0[\Omega] + H_{\text{G}}[\Omega]\hat{q}_{\text{G}}[\Omega] \\
\hat{p}_{\text{out}}[\Omega] &= H_0[\Omega]\hat{p}_0[\Omega] - H_{\text{G}}[\Omega]\hat{p}_{\text{G}}[\Omega]
\end{aligned}
\tag{14}
$$

Assuming that the in-coupled and ancillary modes are in uncorrelated vacuum states, i.e., $\bar{S}_{qq}^0 = \bar{S}_{pp}^0 = \bar{S}_{qq}^{\text{G}} = \bar{S}_{pp}^{\text{G}} = \frac{1}{2}$ and all cross-spectra are identically zero [Supplementary Note I], the output quadrature spectra are $\bar{S}_{qq}^{\text{out}}[\Omega] = \bar{S}_{pp}^{\text{out}}[\Omega] = \frac{1}{2}(|H_0^2[\Omega]| + |H_{\text{G}}^2[\Omega]|)$. Using the explicit forms of the transfer functions [eq. (12)]

$$\bar{S}_{qq}^{\text{out}}[\Omega] = \bar{S}_{pp}^{\text{out}}[\Omega] = \frac{(\sqrt{\eta} - 1/\sqrt{\eta})^2}{4\cos^2(\Omega\tau/2)} + \frac{1}{2}. \tag{15}$$

The frequencies $\Omega_n := (2n+1)\pi/\tau$ (for some integer $n$) at which these spectra are singular are the poles of the transfer functions $H_0, H_{\text{G}}$, and are therefore the frequencies of steady-state oscillations of the closed loop. Physically, they correspond to constructive interference after subsequent traversals of the feedback loop. In a practical oscillator, the gain $G[\Omega]$ is typically engineered to sustain only one steady state at, say, frequency $\Omega_0$ as in Fig. 1D. Then the output field has a single carrier of amplitude $\alpha = \sqrt{\eta}\alpha_{\text{ss}}$ that leaks out of the loop. The abovementioned quadratures then represent fluctuations around this carrier. Defining a frequency offset $\omega$ from the carrier, i.e., $\Omega = \Omega_0 + \omega$ where $\omega\tau \ll 1$, the phase quadrature spectrum in eq. (15) assumes the approximate form

$$\bar{S}_{pp}^{\text{out}}[\Omega_0 + \omega] \approx \frac{(\sqrt{\eta} - 1/\sqrt{\eta})^2}{(\omega\tau)^2} + \frac{1}{2}. \tag{16}$$

The first term is the near-carrier Lorentzian-shaped vacuum noise of the oscillator's output while the second term is the away-from-carrier white vacuum noise.

As it happens, the near-carrier noise described by the first term in eq. (16) is equivalent to the Schawlow-Townes formula. To see this, we describe the phase fluctuation of the output field by the operator[43,44], $\hat{\varphi} \approx \hat{p}_{\text{out}}/(\sqrt{2}|\alpha|)$. (Note that it is the large photon flux of the output field $|\alpha|^2$ that in this instance circumvents the technical difficulties in constructing a hermitian phase operator[45,46]). Thus the frequency spectrum of the output around the carrier is $\bar{S}_{\hat{\varphi}\hat{\varphi}}[\Omega_0 + \omega] = \omega^2 \bar{S}_{\varphi\varphi}[\Omega_0 + \omega] = (\omega^2/2|\alpha|^2)\bar{S}_{pp}^{\text{out}}[\Omega_0 + \omega]$. Using eq. (16),

this is

$$\bar{S}_{\dot{\phi}\dot{\phi}}^{\text{ST}}[\Omega_0 + \omega] \approx \frac{(\sqrt{\eta} - 1/\sqrt{\eta})^2}{2\tau^2|\alpha|^2} + \frac{(\omega\tau)^2}{4\tau^2|\alpha|^2} \approx \frac{(1-\eta)^2}{2\tau^2|\alpha|^2}. \qquad (17)$$

Here we have omitted the second term, which arises from the frequency-independent vacuum noise "$\frac{1}{2}$" in eq. (16), and is negligible close to the carrier. The resulting expression is the Schawlow-Townes formula in its regime of applicability (a laser with a highly reflective out-coupler, i.e., $\eta \approx 1$, in which case eqs. (17) and (1) agree to $\mathcal{O}[(1-\eta)^3]$). Figure 2 compares a quantum noise-limited oscillator's exact output phase spectrum to the Schawlow-Townes formula. Clearly, the Schawlow-Townes formula for a quantum-limited feedback oscillator, which we henceforth take to be eq. (17), arises from quantum vacuum noises added by the in-loop amplifier and the out-coupler.

## Standard quantum limit for feedback oscillators

We will now derive a trade-off between the phase and amplitude fluctuations of the output of a feedback oscillator, contextualize the Schawlow-Townes formula within it, illustrate how squeezed fields can help evade the Schawlow-Townes limit for the phase at the expense of increased power fluctuations, and finally, how entangled fields or a phase-sensitive in-loop amplifier can circumvent the trade-off altogether.

The output field quadratures satisfy the canonical commutation relations $[\hat{q}_{\text{out}}(t), \hat{p}_{\text{out}}(t')] = i\,\delta(t - t')$. This fact alone implies that (see Theorem 1 in "Methods")

$$\bar{S}_{qq}^{\text{out}}[\Omega]\bar{S}_{pp}^{\text{out}}[\Omega] \geq \frac{1}{4}; \qquad (18)$$

a Heisenberg uncertainty principle for the spectra of these quadratures. This constraint is independent of the physics of a feedback oscillator and is thus too lax.

To derive a tighter constraint, we use eq. (14) to relate the output spectra to the in-coupled and ancillary spectra. For each quadrature $x = \{q, p\}$,

$$\begin{aligned}\bar{S}_{xx}^{\text{out}} &= |H_0|^2\bar{S}_{xx}^0 + |H_\text{G}|^2\bar{S}_{xx}^\text{G} \pm 2\,\text{Re}\left[H_0^* H_\text{G}\bar{S}_{xx}^{0,\text{G}}\right] \\ &\geq 2|H_0 H_\text{G}|\left(\left(\sqrt{\bar{S}_{xx}^0\bar{S}_{xx}^\text{G}} - |\bar{S}_{xx}^{0,\text{G}}|\right)\right),\end{aligned} \qquad (19)$$

where we have used the notation $\bar{S}_{qq}^{0,\text{G}} \equiv \bar{S}_{q_0 q_\text{G}}$ etc. The inequality follows from the fact that $a + b \geq 2\sqrt{ab}$, valid for any $a, b \geq 0$, and that $-|z| \leq \text{Re}(z) \leq |z|$ for any complex number $z$. In the first line of eq. (19), the plus sign applies for the amplitude quadrature and the negative sign for the phase quadrature. However, the inequality applies for both quadratures. Using eq. (19), we find that the output quadrature spectra satisfy the constraint

$$\bar{S}_{qq}^{\text{out}}\bar{S}_{pp}^{\text{out}} \geq 4|H_0 H_\text{G}|^2\left(\sqrt{\bar{S}_{qq}^0\bar{S}_{qq}^\text{G}} - |\bar{S}_{qq}^{0,\text{G}}|\right)\left(\sqrt{\bar{S}_{pp}^0\bar{S}_{pp}^\text{G}} - |\bar{S}_{pp}^{0,\text{G}}|\right), \qquad (20)$$

which applies to all phase-insensitive feedback oscillators.

If we assume the modes $\hat{a}_0$ and $\hat{a}_\text{G}$ are uncorrelated, the cross-terms $\bar{S}_{qq}^{0,\text{G}}$ and $\bar{S}_{pp}^{0,\text{G}}$ vanish. Because the modes $\hat{a}_0$ and $\hat{a}_\text{G}$ satisfy the canonical commutation relations, it follows that their quadratures satisfy the uncertainty principle (see Theorem 1 in "Methods") $\bar{S}_{qq}^i\bar{S}_{pp}^i \geq \frac{1}{4}$ for each mode $i = \{0, \text{G}\}$. Thus eq. (20) reduces to

$$\bar{S}_{qq}^{\text{out}}\bar{S}_{pp}^{\text{out}} \geq |H_0 H_\text{G}|^2 = |H_0|^2(|H_0|^2 - 1). \qquad (21)$$

This is a state-independent constraint on the fluctuations in the output field of a feedback oscillator formed by positive feedback of a phase-insensitive amplifier in the absence of quantum correlations between its in-coupled and ancillary modes.

We take eq. (21) to be a form of a standard quantum limit (SQL) for the output of a feedback oscillator ("standard" here meaning the setting where no additional quantum strategies are employed). Around the carrier, where $|H_0| \gg 1$, eq. (21) is much tighter than the Heisenberg uncertainty principle in eq. (18). Indeed, the Schawlow-Townes limit is an instance of eq. (21). To wit, note that for frequencies near the carrier, $|H_0| \gg 1$, so eq. (21) implies $\bar{S}_{qq}^{\text{out}}\bar{S}_{pp}^{\text{out}} \gtrsim |H_0|^4$. The Schawlow-Townes limit is the case where this inequality is saturated by an equal partitioning of fluctuations between the two output quadratures, i.e., $\bar{S}_{qq}^{\text{out,ST}} \approx \bar{S}_{pp}^{\text{out,ST}} \approx |H_0|^2$. Thus, for a feedback oscillator with a linear amplifier and linear output coupler, as long as the modes $\hat{a}_0$ and $\hat{a}_\text{G}$ are independent, and the amplifier is phase-insensitive, any attempt to reduce frequency fluctuations below the Schawlow-Townes limit—by engineering the out-coupler or ancillary states—will elicit increased fluctuations in the output power of the oscillator. Any and all of these assumptions can be relaxed to evade the SQL with varying degrees of malleability.

## Phase-insensitive in-loop amplifier: squeezing and entanglement

It is apparent that the bound in eq. (20) can be weakened by correlating the modes $\hat{a}_0, \hat{a}_\text{G}$. To explicate this, we focus on the quadrature fluctuations near the carrier by expressing eq. (14) at the offset frequency $\Omega_0 + \omega$

$$\begin{aligned}\hat{q}_{\text{out}}[\omega] &\approx H_0[\omega]\left(\hat{q}_0[\omega] - \hat{q}_\text{G}[\omega]\right) \\ \hat{p}_{\text{out}}[\omega] &\approx H_0[\omega]\left(\hat{p}_0[\omega] + \hat{p}_\text{G}[\omega]\right).\end{aligned} \qquad (22)$$

For brevity, here and henceforth, we write $\omega$ in lieu of $\Omega_0 + \omega$. Then the spectrum of the output phase and amplitude quadratures assumes the general form

$$\begin{aligned}\bar{S}_{qq}^{\text{out}}[\omega] &= |H_0[\omega]|^2\left(\bar{S}_{qq}^0[\omega] + \bar{S}_{qq}^\text{G}[\omega] - 2\,\text{Re}\left[\bar{S}_{qq}^{0,\text{G}}[\omega]\right]\right) \\ \bar{S}_{pp}^{\text{out}}[\omega] &= |H_0[\omega]|^2\left(\bar{S}_{pp}^0[\omega] + \bar{S}_{pp}^\text{G}[\omega] + 2\,\text{Re}\left[\bar{S}_{pp}^{0,\text{G}}[\omega]\right]\right).\end{aligned} \qquad (23)$$

Clearly, squeezing either the in-coupled or ancillary mode can reduce the noise power in a desired quadrature. Entangling these modes—resulting in non-zero $\bar{S}_{qq}^{0,\text{G}}$ and $\bar{S}_{pp}^{0,\text{G}}$—can reduce fluctuations in both quadratures simultaneously. For illustration, we imagine frequency-independent single-mode squeezing of the in-coupled mode $\hat{a}_0$ and the amplifier's ancillary mode $\hat{a}_\text{G}$, with squeezing parameters $r_0$ and $r_\text{G}$ respectively, followed by two-mode squeezing of the modes $\hat{a}_0$ and $\hat{a}_\text{G}$ with squeezing parameter $r_\text{E}$. (The latter corresponds to continuous-variable EPR entanglement of the two modes $\hat{a}_0$ and $\hat{a}_\text{G}$[47].) Gaussian state techniques allow the resulting output spectra to be derived (see Supplementary Note I for details):

$$\begin{aligned}\bar{S}_{qq}^{\text{out}}[\omega] &= \frac{1}{2}e^{-r_\text{E}}\left(e^{2r_0} + e^{2r_\text{G}}\right)\bar{S}_{qq}^{\text{out,ST}}[\omega] \\ \bar{S}_{pp}^{\text{out}}[\omega] &= \frac{1}{2}e^{-r_\text{E}}\left(e^{-2r_0} + e^{-2r_\text{G}}\right)\bar{S}_{pp}^{\text{out,ST}}[\omega]\end{aligned} \qquad (24)$$

By squeezing their phase-quadratures, corresponding to $r_0, r_\text{G} > 0$, we can suppress the oscillator's phase quadrature fluctuations—and therefore the linewidth of the oscillator—below the Schawlow-Townes limit without bound, at the expense of increasing the oscillator's amplitude quadrature fluctuations. By correlating these modes, corresponding to $r_\text{E} > 0$, we can simultaneously reduce the oscillator's amplitude and phase quadrature fluctuations below the Schawlow-

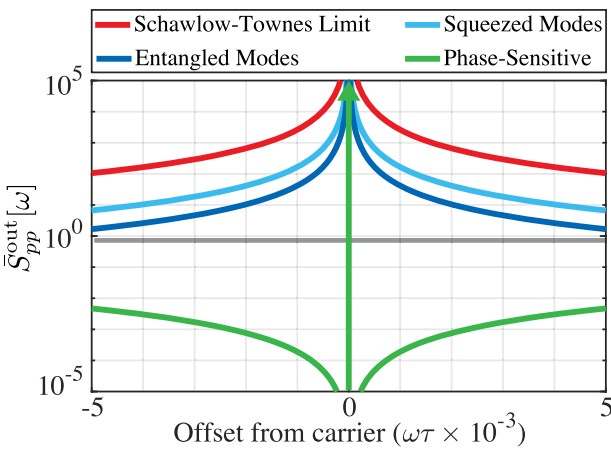

**Fig. 3 | Phase noise of quantum-enhanced feedback oscillators.** Spectra of the output phase quadrature for four types of quantum noise-limited oscillators. Red shows the Schawlow-Townes spectrum of an oscillator with phase-insensitive amplifier and the in-coupled and ancillary modes in vacuum. Light and darker blues depict the case where these modes are squeezed (light blue) and entangled (dark blue) (both with 12 dB of squeezing). Green shows the case where the in-loop amplifier is purely phase-sensitive. Gray shows $\bar{S}_{pp}^{\text{out}} = 1/2$ for reference.

Townes limit. However, the Heisenberg uncertainty relation [eq. (18)] $\bar{S}_{qq}^{\text{out}} \bar{S}_{pp}^{\text{out}} \geq \frac{1}{4}$ still holds and represents the limit to which noise in the output quadratures of a feedback oscillator can be suppressed simultaneously (see Supplementary Note I for an explicit verification of this fact for EPR entangled inputs).

These results show that oscillators featuring sub-Schawlow-Townes phase noise performance are possible with linear (phase-insensitive) amplifiers and linear out-couplers but with the injection of squeezed or entangled states into the feedback loop. Recent work[48,49] has shown that sub-Schawlow-Townes performance is also achievable with out-couplers that feature a nonlinear response to the fields that impinge it and that such nonlinear systems need not trade off amplitude stability to realize enhanced phase stability[50,51].

### Phase-sensitive in-loop amplifier

An alternative method of reducing the oscillator's output amplitude and phase spectra is to modify the feedback loop itself by replacing the phase-insensitive amplifier with a phase-sensitive amplifier as depicted in Fig. 1B. This figure shows a general phase-sensitive amplifier decomposed (see "Methods" subsection "Decomposing a phase-sensitive amplifier as a phase-insensitive amplifier and a squeezer") into a phase-insensitive amplifier followed by an ideal (i.e., noiseless) phase-sensitive amplifier with squeezing parameter $\xi = r_s e^{i\varphi_s}$. When the phase-insensitive component has unity-gain, this cascade realizes a noiseless phase-sensitive amplifier. As before, the nonlinear saturating response of the phase-insensitive amplifier limits the oscillator's output; a straightforward extension of the prior analysis shows that this happens when $Ge^{r_s}\sqrt{\eta} = 1$, which can be interpreted as a balance between the phase-sensitive gain ($Ge^{r_s}$) and loss in the loop. Linear response around this steady-state oscillation is described by

$$\hat{a}_{\text{out}}^{-}[\Omega] = \cosh(r_s)\,\hat{a}_s[\Omega] + e^{i\varphi_s}\sinh(r_s)\,\hat{a}_s^{\dagger}[\Omega]$$
$$\hat{a}_s[\Omega] = G\,\hat{a}_{\text{in}}[\Omega] + \sqrt{G^2 - 1}\,\hat{a}_G^{\dagger}[\Omega] \tag{25}$$

to replace the response of the in-loop amplifier in the phase-insensitive case [eq. (9)]. Here, as before, $\hat{a}_0, \hat{a}_G$ are the modes conveying noise at the out-coupler and the in-loop amplifier. The full set of loop equations can be solved as before, and it can be shown that the imposition of

canonical commutation relations on the in-coupled and outgoing fields imply that the ancillary mode $\hat{a}_G$ also obeys them (see Supplementary Note III A for details).

In the extreme case where the in-loop amplifier is purely phase-sensitive ($G = 1$ in eq. (25)), i.e., a single-mode squeezer with no additional noise, the output quadrature spectra are (see Supplementary Note III for this derivation and the more general phase-sensitive case)

$$\bar{S}_{qq}^{\text{out}}[\Omega] = \left(1 + \frac{(\sqrt{\eta} - 1/\sqrt{\eta})^2}{4\cos^2(\Omega\tau/2)}\right)\bar{S}_{qq}^{0}[\omega]$$
$$\bar{S}_{pp}^{\text{out}}[\Omega] = \left(\frac{4\cos^2(\Omega\tau/2)}{(\sqrt{\eta} - 1/\sqrt{\eta})^2 + 4\cos^2(\Omega\tau/2)}\right)\bar{S}_{pp}^{0}[\omega]. \tag{26}$$

Since $\bar{S}_{qq}^{\text{out}}\bar{S}_{pp}^{\text{out}} = \bar{S}_{qq}^{0}\bar{S}_{pp}^{0}$, the feedback oscillator does not increase the uncertainty product between the amplitude and phase quadratures from its input to output. In particular, if the in-coupled field $\hat{a}_0$ is vacuum, the oscillator's output field is a minimal uncertainty squeezed state around the steady-state oscillating carrier—a bright squeezed state.

In practice, technical phase noise[52] and pump noise in the squeezer will contaminate its output, precluding the zero phase noise around the carrier predicted by eq. (26). Note that the predictions of eq. (26), based on a model of a phase-sensitive amplifier in a feedback loop, is consistent with a Hamiltonian model of an optical parametric oscillator (OPO) (see Supplementary Note IV); the latter elucidates the origin of phase-sensitive amplifier saturation and the conditions necessary to minimize the effects of pump noise.

## Discussion

Figure 3 compares the output phase quadrature spectra of oscillators with squeezed in-coupled and ancillary modes, squeezed and entangled in-coupled and ancillary modes, and an oscillator with a phase-sensitive amplifier. As can be seen in the figure, embedding a phase-sensitive amplifier in the feedback loop suppresses the oscillator's phase-quadrature spectrum much more than is possible by squeezing or correlating the input modes for a given maximal level of squeezing. It is worth noting that, unlike squeezing the in-coupled and ancillary modes, entangling these modes or embedding a phase-sensitive amplifier in the feedback loop reduces the oscillator's output phase noise without increasing amplitude noise.

We have identified the origin of the Schawlow-Townes limit to the frequency stability of an oscillator in a manner that is applicable to a wide class of feedback oscillators. In fact, for a phase-insensitive, quantum-noise-limited oscillator, it is one facet of a more general SQL for the oscillator's outgoing field. This SQL dictates the fundamental trade-off between the frequency and power fluctuations of a broad class of feedback oscillators. However, systematic strategies such as injection of squeezed vacuum, EPR entanglement, and phase-sensitive amplification offer sufficient room to evade the Schawlow-Townes limit and thereby realize a qualitatively new class of stable oscillators.

## Methods

### Spectral densities and their uncertainty principle

For a time-dependent operator (not necessarily hermitian) $\hat{A}(t)$, we define its Fourier transform as

$$\hat{A}[\Omega] = \int_{-\infty}^{+\infty} \hat{A}(t)e^{i\Omega t}\,\mathrm{d}t, \tag{27}$$

where $\Omega \in \mathbb{R}$. Note that the Fourier transform of the hermitian conjugate, which we denote by $\hat{A}^{\dagger}[\Omega]$, is different from the hermitian conjugate of the Fourier transform, which we represent by $\hat{A}[\Omega]^{\dagger}$; the

two are related as,

$$\hat{A}^{\dagger}[\Omega] = \hat{A}[-\Omega]^{\dagger}. \tag{28}$$

If $\hat{A}$ is hermitian, i.e., $\hat{A}(t)^{\dagger} = \hat{A}(t)$, then

$$\hat{A}^{\dagger}[\Omega] = \hat{A}[\Omega]. \tag{29}$$

The inverse of eq. (27) is given by,

$$\hat{A}(t) = \int_{-\infty}^{+\infty} \hat{A}[\Omega] e^{-i\Omega t} \frac{d\Omega}{2\pi}. \tag{30}$$

We define the cross-correlation between the two general (not necessarily hermitian and not necessarily commuting) operators $\hat{A},\hat{B}$ by the symmetrized expression,

$$\bar{S}_{AB}(t) = \frac{1}{2}\left\langle \{\hat{A}^{\dagger}(t),\hat{B}(0)\}\right\rangle.$$

We will exclusively use the symmetrized double-sided cross-correlation spectrum, defined as the Fourier transform of the symmetrized cross-correlation:

$$\begin{aligned}\bar{S}_{AB}[\Omega] &= \int_{-\infty}^{\infty} \frac{1}{2}\left\langle \{\hat{A}^{\dagger}(t),\hat{B}(0)\}\right\rangle e^{i\Omega t}\, dt \\ &= \int_{-\infty}^{\infty} \frac{1}{2}\left\langle \{\hat{A}^{\dagger}[\Omega],\hat{B}[\Omega']\}\right\rangle \frac{d\Omega'}{2\pi},\end{aligned} \tag{31}$$

where the last equality follows from using the Fourier representation [eq. (30)] of the time-dependent operators.

For weak-stationary operators, i.e., those that pair-wise satisfy $\langle\hat{A}(t)\hat{B}(t')\rangle = \langle\hat{A}(t-t')\hat{B}(0)\rangle$, the spectrum is given by the identity,

$$\bar{S}_{AB}[\Omega] \cdot 2\pi\delta[\Omega+\Omega'] = \frac{1}{2}\left\langle \{\hat{A}^{\dagger}[\Omega],\hat{B}[\Omega']\}\right\rangle. \tag{32}$$

Note that in this case, $\bar{S}_{AB}[\Omega]^{*} = \bar{S}_{BA}[\Omega]$.

**Lemma 1.** The spectrum of a weak-stationary (but not necessarily hermitian) operator is positive; i.e.,

$$\bar{S}_{AA}[\Omega] \geq 0. \tag{33}$$

if $\langle\hat{A}^{\dagger}(t)\hat{A}(t')\rangle = \langle\hat{A}^{\dagger}(t-t')\hat{A}(0)\rangle$.

**Proof.** Since $\hat{A}$ is weak-stationary, eq. (32), together with eq. (28), implies that

$$\bar{S}_{AA}[\Omega] \cdot 2\pi\delta[0] = \frac{1}{2}\left\langle \{\hat{A}[-\Omega]^{\dagger},\hat{A}[-\Omega]\}\right\rangle.$$

Next, we prove the right-hand side is positive. Consider the first term, which is the expectation of the hermitian operator, $\hat{A}[-\Omega]^{\dagger}\hat{A}[-\Omega]$ over some state, say $\hat{\rho}$. Since the general state $\hat{\rho}$ can be expressed as a convex combination, $\hat{\rho} = \sum_{i} p_i |\psi_i\rangle\langle\psi_i|$, with $p_i \geq 0, \sum_i p_i = 1$, and $\langle\psi_i|\psi_j\rangle = \delta_{ij}$, the expectation value may be written as,

$$\begin{aligned}\left\langle\hat{A}[-\Omega]^{\dagger}\hat{A}[-\Omega]\right\rangle &= \mathrm{Tr}\left[\hat{A}[-\Omega]^{\dagger}\hat{A}[-\Omega]\hat{\rho}\right] \\ &= \sum_i p_i\langle\psi_i|\hat{A}[-\Omega]^{\dagger}\hat{A}[-\Omega]|\psi_i\rangle \\ &= \sum_i p_i \|\hat{A}[-\Omega]|\psi_i\rangle\|^2 \geq 0.\end{aligned}$$

The same argument applies to the second term. $\square$

**Theorem 1.** (Uncertainty principle for spectra) For weak-stationary hermitian operators $\hat{A}$ and $\hat{B}$ that satisfy the commutation relationship $[\hat{A}[\Omega],\hat{B}[\Omega']] = ic \cdot 2\pi\delta[\Omega+\Omega']$ for some real constant $c$,

$$\bar{S}_{AA}[\Omega]\bar{S}_{BB}[\Omega] \geq \frac{c^2}{4} + \left|\bar{S}_{AB}[\Omega]\right|^2. \tag{34}$$

**Proof.** Let $\{\hat{A}_j\}$ be a set of weak-stationary observables; then their linear combination, $\hat{M} = \sum_j \alpha_j \hat{A}_j$ with $\alpha_j \in \mathbb{C}$ is also weak-stationary. Using the fact that $\langle\hat{M}[-\Omega]^{\dagger}\hat{M}[-\Omega]\rangle \geq 0$ as shown in the proof of Lemma 1, we have

$$\sum_{j,k} \alpha_j^{*}\alpha_k \left\langle\hat{A}_j[\Omega]\hat{A}_k[-\Omega]\right\rangle \geq 0,$$

where we have used eq. (29). Splitting $\hat{A}_j[\Omega]\hat{A}_k[-\Omega]$ into Hermitian and anti-Hermitian parts

$$\hat{A}_j[\Omega]\hat{A}_k[-\Omega] = \frac{1}{2}\left\{\hat{A}_j[\Omega],\hat{A}_k[-\Omega]\right\} + \frac{1}{2}\left[\hat{A}_j[\Omega],\hat{A}_k[-\Omega]\right],$$

the above inequality becomes

$$\sum_{j,k} \alpha_j^{*}\alpha_k \left(\bar{S}_{A_j A_k}[\Omega] + \bar{C}_{A_j A_k}[\Omega]\right) \geq 0, \tag{35}$$

where

$$\bar{C}_{AB}[\Omega] \equiv \int_{-\infty}^{\infty} \frac{1}{2}\left\langle\left[\hat{A}^{\dagger}[\Omega],\hat{B}[\Omega']\right]\right\rangle \frac{d\Omega'}{2\pi}.$$

Since the inequality in eq. (35) is true for arbitrary $\alpha_j$, it requires that the eigenvalues of the matrix with elements $\bar{S}_{A_j A_k}[\Omega] + \bar{C}_{A_j A_k}[\Omega]$ be non-negative. Specifically, the smallest eigenvalue of this matrix must be non-negative.

Consider now the case where $\{\hat{A}_j\} = \{\hat{A},\hat{B}\}$ with $[\hat{A}[\Omega],\hat{B}[\Omega']] = ic \cdot 2\pi\delta[\Omega+\Omega']$. Equation (35) then requires

$$\bar{S}_{AA}\bar{S}_{BB} - \left(\bar{S}_{AB}+\bar{C}_{AB}\right)\left(\bar{S}_{BA}+\bar{C}_{BA}\right) \geq 0.$$

Simplifying using the fact that $\bar{C}_{AB}[\Omega] = \bar{C}_{BA}^{*}[\Omega] = ic/2$ and $\bar{S}_{AB}[\Omega] = \bar{S}_{BA}^{*}[\Omega]$, gives eq. (34). $\square$

## Saturating behavior of phase-insensitive feedback oscillators and classical steady state

Here, we analyze the classical steady-state behavior of the saturating feedback oscillator shown in Fig. 1A. Classically, the behavior of the system shown in Fig. 1A is governed in the time domain by

$$\begin{aligned}\alpha_{\mathrm{out}}^{-}(t) &= \mathcal{A}[\alpha_{\mathrm{in}}(t)] \\ \alpha_{\mathrm{out}}^{+}(t) &= -\sqrt{\eta}\,\alpha_{\mathrm{out}}^{-}(t) + \sqrt{1-\eta}\,\alpha_0(t) \\ \alpha_{\mathrm{out}}(t) &= \sqrt{1-\eta}\,\alpha_{\mathrm{out}}^{-}(t) + \sqrt{\eta}\,\alpha_0(t) \\ \alpha_{\mathrm{in}}(t) &= \alpha_{\mathrm{out}}^{+}(t-\tau).\end{aligned} \tag{36}$$

where $\{\alpha_i\}$ are classical field amplitudes and $\mathcal{A}[\cdot]$ is the amplifier's nonlinear response. Equation (36) can be simplified to

$$\alpha_{\mathrm{out}}^{+}(t) = -\sqrt{\eta}\,\mathcal{A}[\alpha_{\mathrm{out}}^{+}(t-\tau)] + \sqrt{1-\eta}\,\alpha_0(t-\tau) \tag{37}$$

The input $\alpha_0(t)$ sources the classical field that circulates in the loop. In reality, the input represented by $\alpha_0$ is pure noise, in the ideal case, just vacuum noise.

To analyze how the feedback loop attains a steady state by being driven purely by noise, we assume that $\alpha_0$ represents infinitesimally

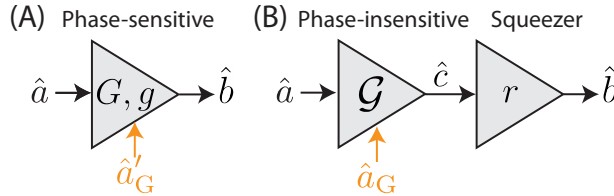

**Fig. 4 | Phase-sensitive amplifier decomposition. A** A phase-sensitive amplifier. **B** A decomposition of the phase-sensitive amplifier from (**A**) into a phase-insensitive amplifier followed by a squeezer. Noise modes are shown in orange.

small fluctuations. In order to understand how the loop starts, consider that $\alpha_0(0)$ is a small random value and zero for $t > 0$. Let this produce a small amplitude $\alpha_{\text{out}}^+(0 < t < \tau) = \delta$. We are primarily interested in the circulating power rather than phase rotations, so taking the magnitude square of eq. (37) under these conditions:

$$|\alpha_{\text{out}}^+(t)|^2 = \eta \left| \mathcal{A}[\alpha_{\text{out}}^+(t - \tau)] \right|^2$$
$$\alpha_{\text{out}}^+(0 < t < \tau) = \delta. \tag{38}$$

That is, if the initial random seed $\alpha_0(0)$ produces a steady state amplitude, it must satisfy

$$|\mathcal{A}[\alpha_{\text{out}}^+]| = \alpha_{\text{out}}^+/\sqrt{\eta}, \tag{39}$$

which defines the steady state $\alpha_{ss}$ of eq. (7).

The above is only a necessary condition since the question of the stability of this steady state remains open. We will now show that the properties of $\mathcal{A}$ listed in the "saturation, steady-state, and linear gain" section suffice to ensure stability. Properties (1), (3), and (4) imply that $\mathcal{A}'(x) = \frac{dA}{dx} < \frac{A(x)}{x} \quad \forall x$. Let $\alpha_{\text{out}}^+(t) = \alpha_{ss} + \delta_t$ where $\delta_t$ is a small perturbation to the steady state value of $\alpha_{\text{out}}^+$ at time $t$. Similarly, let $\alpha_{\text{out}}^+(t + \tau) = \alpha_{ss} + \delta_\tau$. Equation (38) then reads $|\alpha_{ss} + \delta_\tau|^2 = \eta |\mathcal{A}[\alpha_{ss} + \delta_t]|^2$. Expanding to first order in $\delta_t, \delta_\tau$ and using eq. (39) gives

$$|\delta_\tau| = \frac{\alpha_{ss} \mathcal{A}'[\alpha_{ss}]}{\mathcal{A}[\alpha_{ss}]} |\delta_t|$$
$$\Rightarrow \quad |\delta_\tau| < |\delta_t| \tag{40}$$

where we have used the properties of $\mathcal{A}$. Thus, perturbations around the steady state tend to die over time; i.e., the steady state is stable to small perturbations.

Similarly, we can show that the operating point where all amplitudes are zero is unstable. That is, any initial fluctuation will drive the loop to its steady state eq. (39). Now, let $\alpha_{\text{out}}^+(t) = \delta_t$ and let $\alpha_{\text{out}}^+(t + \tau) = \delta_\tau$ where $\delta_t$ and $\delta_\tau$ are sufficiently small that property (3) is satisfied. We have

$$|\delta_\tau|^2 = \eta |\mathcal{A}[\delta_t]|^2$$
$$\Rightarrow \quad |\delta_\tau| = \sqrt{\eta} |G_0| |\delta_t| \tag{41}$$
$$\Rightarrow \quad |\delta_\tau| > |\delta_t|.$$

Here $G_0 = \mathcal{A}'[0]$, the linear slope of the amplifier's nonlinear response around zero.

In sum, the loop will reach a steady state with a large circulating classical field with amplitude $\alpha_{ss}$ and a linearized gain of

$$G \equiv \lim_{t \to +\infty} \frac{\alpha_{\text{out}}^-(t)}{\alpha_{\text{in}}(t)} = \frac{\mathcal{A}[\alpha_{ss}]}{\alpha_{ss}} = \frac{1}{\sqrt{\eta}}. \tag{42}$$

We emphasize two key points of this classical analysis of saturating feedback oscillators. First, any small initial fluctuation will be amplified, eventually reaching a stable equilibrium with a large-

amplitude output field $\alpha_{ss}$ circulating in the loop. This oscillating field is the fundamental feature of a positive feedback oscillator. Second, at this equilibrium point, the gain medium is linear for small perturbations around the large field, with a gain given by the requirement for the system to conserve round-trip power in steady-state, i.e., $G = 1/\sqrt{\eta}$. The fact that any amplifier satisfying properties (1) through (4) embedded in a feedback loop saturates to a point where it can be treated as having a linear gain allows for a linear quantum mechanical analysis of the oscillator's phase and amplitude fluctuations.

### Linear response of phase-insensitive feedback oscillators

Consider the positive feedback amplifier configuration shown in Fig. 1A. The output of a phase-insensitive amplifier with gain $G$ is coupled back into its input after attenuation by a factor $\sqrt{\eta}$ and a delay of $\tau$. The remaining fraction of the signal is coupled out of the loop to derive the out-of-loop field $a_{\text{out}}$.

The equations of motion for the system are obtained by going around the loop in Fig. 1A. For the Heisenberg-picture operators in the time domain, we have

$$\hat{a}_{\text{out}}^-(t) = G\,\hat{a}_{\text{in}}(t) + \sqrt{G^2 - 1}\,\hat{a}_{\text{G}}^\dagger(t)$$
$$\hat{a}_{\text{out}}^+(t) = -\sqrt{\eta}\,\hat{a}_{\text{out}}^-(t) + \sqrt{1 - \eta}\,\hat{a}_0(t)$$
$$\hat{a}_{\text{out}}(t) = \sqrt{1 - \eta}\,\hat{a}_{\text{out}}^-(t) + \sqrt{\eta}\,\hat{a}_0(t) \tag{43}$$
$$\hat{a}_{\text{in}}(t) = \hat{a}_{\text{out}}^+(t - \tau)$$

The ancillary mode $\hat{a}_{\text{G}}$ describes the unavoidable noise added in any phase-insensitive linear amplifier[36,37].

Equation (43) can be solved in the frequency domain for the output field $\hat{a}_{\text{out}}[\Omega]$ in terms of the inputs $\hat{a}_0[\Omega]$ and $\hat{a}_{\text{G}}[\Omega]$. The result is

$$\hat{a}_{\text{out}}[\Omega] = \frac{\sqrt{\eta} + Ge^{i\Omega\tau}}{1 + G\sqrt{\eta}e^{i\Omega\tau}}\hat{a}_0[\Omega] + \frac{\sqrt{G^2 - 1}\sqrt{1 - \eta}}{1 + G\sqrt{\eta}e^{i\Omega\tau}}\hat{a}_{\text{G}}^\dagger[\Omega]$$
$$\equiv H_0[\Omega]\hat{a}_0[\Omega] + H_{\text{G}}[\Omega]\hat{a}_{\text{G}}^\dagger[\Omega], \tag{44}$$

where $H_{0,\text{G}}$ are the corresponding linear response transfer functions. In steady state, $G = 1/\sqrt{\eta}$ as discussed in the "Methods" section "saturating behavior of phase-insensitive feedback oscillators and classical steady state", and so the feedback path is characterized by two quantities, the beam-splitter transmissivity $\eta$ and the delay $\tau$; so

$$H_0[\Omega] = \frac{\sqrt{\eta} + \frac{1}{\sqrt{\eta}}e^{i\Omega\tau}}{1 + e^{i\Omega\tau}}, \quad H_{\text{G}}[\Omega] = \frac{1/\sqrt{\eta} - \sqrt{\eta}}{1 + e^{i\Omega\tau}}. \tag{45}$$

### Decomposing a phase-sensitive amplifier as a phase-insensitive amplifier and a squeezer

In this section, we show that a quantum-limited phase-sensitive amplifier can always be decomposed into an ideal phase-insensitive amplifier followed by an ideal single-mode squeezer.

Consider a phase-sensitive amplifier that amplifies the input $\hat{a}$ to produce the output $\hat{b} = G\hat{a} + g\hat{a}^\dagger + \hat{a}_{\text{G}}^\dagger$, where $\hat{a}_{\text{G}}$ is the amplifier's ancillary mode (see Fig. 4A). (Note that the ancillary mode, $\hat{a}_{\text{G}}$, does not necessarily have bosonic statistics here.) We assert that such a phase-sensitive amplifier is equivalent to an ideal phase-insensitive amplifier with gain $\mathcal{G} = \sqrt{G^2 - g^2}$ that sends mode $\hat{a}$ to the mode $\hat{c} = \mathcal{G}\hat{a} + \hat{a}_{\text{G}}^{\prime\dagger}$, followed by an ideal squeezer which sends the mode $\hat{c}$ to the output mode $\hat{b} = \cosh(r)\hat{c} + \sinh(r)\hat{c}^\dagger$ with $\tanh(r) = g/G$ (see Fig. 4B). For this equivalence to hold, we require $\hat{a}_{\text{G}}^{\prime\dagger} = (G\hat{a}_{\text{G}}^\dagger + g\hat{a}_{\text{G}}^\prime)/\mathcal{G}$.

We can prove this by explicit computation. In the case of the phase-sensitive amplifier, the output mode $\hat{b}$ is given in terms of the input mode $\hat{a}$ by $\hat{b} = G\hat{a} + g\hat{a}^\dagger + \hat{a}_{\text{G}}^\dagger$. In the case of a phase-insensitive

amplifier followed by a squeezer, we have (assuming $G$ and $g$ are positive)

$$\begin{aligned}
\hat{b} &= \cosh(r)\hat{c} + \sinh(r)\hat{c}^\dagger \\
&= \cosh(r)(\mathcal{G}\hat{a} + \hat{a}'^\dagger_{\mathrm{G}}) + \sinh(r)(\mathcal{G}\hat{a}^\dagger + \hat{a}'_{\mathrm{G}}) \\
&= G\hat{a} + g\hat{a}^\dagger + \frac{G\hat{a}'^\dagger_{\mathrm{G}} + g\hat{a}'_{\mathrm{G}}}{\sqrt{G^2 - g^2}} \\
&\equiv G\hat{a} + g\hat{a}^\dagger + \hat{a}^\dagger_{\mathrm{G}},
\end{aligned} \qquad (46)$$

Additionally, we can show that $\hat{a}_{\mathrm{G}}$ and $\hat{a}'_{\mathrm{G}}$ have the same commutation relations.

$$\begin{aligned}
\left[ a_{\mathrm{G}}, a^\dagger_{\mathrm{G}} \right] &= \frac{\left[ Ga'_{\mathrm{G}} + ga'^\dagger_{\mathrm{G}}, Ga'^\dagger_{\mathrm{G}} + ga'_{\mathrm{G}} \right]}{G^2 - g^2} \\
&= \frac{G^2 \left[ a'_{\mathrm{G}}, a'^\dagger_{\mathrm{G}} \right] + g^2 \left[ a'^\dagger_{\mathrm{G}}, a'_{\mathrm{G}} \right]}{G^2 - g^2} \\
&= \left[ a'_{\mathrm{G}}, a'^\dagger_{\mathrm{G}} \right],
\end{aligned} \qquad (47)$$

so we see that this is the correct decomposition of a phase-sensitive amplifier.

The primary benefit of this decomposition is that it clarifies the extent to which a phase-sensitive amplifier must add noise: only the phase-insensitive component in its decomposition adds noise, while the squeezer is noiseless.

## Data availability
Figure 2 is a straightforward plot of eq. (15) and (16), while Fig. 3 is a straightforward plot of eqs. (15), (24) and (26) respectively. Supplementary Figure 1 is a straightforward plot of Supplementary Equation (7). The relevant computer code is available upon request to the authors.

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

## Acknowledgements

H.A.L. gratefully acknowledges the support of the National Science Foundation through the LIGO operations cooperative agreement PHY-18671764464.

## Author contributions

V.S. conceived the project and guided H.A.L. through it. The authors jointly developed the results and wrote the manuscript.

## Competing interests

US provisional patent 63/477,416 ("Methods of Enhancing the Stability of Quantum Noise Limited Feedback Oscillators") has been jointly filed by H.A.L. and V.S. based on some of the theories presented here.
