## [Peer Review File · Nature Communications]

Quantum noise and its evasion in feedback oscillatorsREVIEWER COMMENTS

Reviewer #1 (Remarks to the Author):

The manuscript provides a simple but self-consistent model to study the fundamental output noise of a feedback oscillator. In spite of the simplicity, the authors recover the well known Schawlow-Townes (ST) formula in the semi-classical regime, and identify the extra components due to shot noise as well as quantum correlations. The simple model does not only provide new understanding of the origin of the ST formula, but also allows the authors to develop quantum strategies to suppress the oscillator noise, i.e. by using squeezing and entanglement. Although I am not conversant with all the literature about feedback oscillator noise, the results presented in the manuscript appear to be novel and generic, and can be applied to analyze a wide range of feedback oscillator in the quantum regime.

Overall, I enjoy reading the manuscript, and most result is solid to me. I believe it is a noteworthy result for, as the authors suggested, quantum electronics and optics communities and beyond, and can have an impact in building quantum devices for metrology, communication, etc. However, there are couple of areas that I would like the authors to improve or clarify before I recommend it for publication.

1. The current version of manuscript is focused on the techniques, but the discussion of its real world connection is little. I suggest the authors to include more discussion on the practical side, including possible experimental setups that can verify the theoretical results, and explicit examples of potential application, e.g. engineering laser with lower phase noise, improving quantum information processors.

2. I appreciate that the authors are deriving the results based on generic model and assumptions. However, while reading I could not stop making comparisons to the physical models of amplifiers I know. This could have subtle implications, as I explain later. Generally, I think it would be helpful to readers like me if the authors can include an example, at least in appendix, to show how the derivations agree with explicit physical models of amplifier (e.g. standard Hamiltonian and coupling of optical parametric oscillator).

Here are some subtle technical comments:

3. At the beginning of p.2, the authors argue that the standard properties of linear amplifier "do not apply when the feedback loop is closed", they substantiate their argument by the fact that the gain will saturate. I do not fully agree with this argument. If one has a strictly linear amplifier, as the assumption of first paragraph of Sec. I, one can still construct such a feedback loop, except that the gain is exponential increasing and there is no steady state. The gain saturates is not due to the loop but the intrinsic non-linearity of the amplifier that becomes significant at large gain. One could also consider that the gain would saturate if the amplifier is sufficiently strongly driven, without introducing the loop.

4. In Sec. IA, the authors present four generic assumptions of the amplifier. I find the assumption 4 might be too restrictive. I suppose it would be sufficient for the oscillator to reach a steady state if $dA/dx < 0$ for some sufficiently large x ? Assumption 4 seems to be suitable for laser-like system, which the photon output is determined by the fixed total number of atoms, but there could be other amplifiers (e.g. Josephson Parametric Amplifier) that the steady state is determined by the competition of multiple effects.

5. In the phase insensitive model, I could understand that at the steady state the input and output are dominated by a strong displacement. By moving to the frame of the displacement, the leading order Hamiltonian in the amplifier will be quadratic in the mode operators, this gives the linear transformation in first line of Eq. 9. There are still higher-than-quadratic terms, as they must present for the existence of steady state, but their influence on quantum fluctuation is weak in most physical model of amplifier, thus the model in this manuscript is valid. However, the situation might not be the same in the phase sensitive case. It is known that squeezing can amplify the nonlinear interaction (see e.g. doi.org/10.1103/PhysRevLett.120.093601 and doi.org/10.1103/PhysRevLett.120.093602). If we consider the amplifier at strong phase sensitive amplification, in the squeezed frame the higher-than-quadratic terms are not necessarily negligible, so the input-output relation might not be linear. I believe the imperfect phase sensitive amplifier model in this manuscript is still valid if the phase insensitive gain dominates the overall gain, the authors should be more specific about the assumption and discuss its validity in physical systems. On the other hand, the "perfectly phase sensitive" case is doubtful to me. It is not clear if there could be self-consistent model that the higher-than-quadratic terms can both contribute to

the saturation of phase-sensitive gain and not inducing nonlinear transformation of the mode operators.

Reviewer #2 (Remarks to the Author):

Key results:

In the manuscript entitled “Quantum noise and its evasion in feedback oscillators”, the authors consider a feedback oscillator consisting of a phase insensitive amplifier, that is embedded in a feedback loop, where the coupling of this feedback oscillator to the output field is treated using a beam splitter interaction. A linearised analysis of this system is performed, by treating the quantum-mechanical fluctuations of the amplified field as a small perturbation about a mean amplitude obtained from a classical treatment.

From this analysis, they identify the sources that give rise to noise in the output field of the feedback oscillator, which can be traced back to the quantum noises added by the ancillary mode of the amplifier, as well as the vacuum input port of the beam splitter. The formula which is derived for the frequency noise within this system agrees with the Schawlow-Townes formula. Further, an inequality is derived for the product of the spectral variances for two quadratures, corresponding to the real and imaginary parts of the output field amplitude in their linearized treatment. In the special case where there are no correlations between the ancillary modes of the amplifier and beam splitter, this inequality is assigned as the “standard quantum limit”.

In the final sections of the main text, the authors investigate how modifying various aspects of their system influence this inequality. They show that by squeezing the ancillary amplifier mode, or that of the beam splitter, then fluctuations in one of the output field quadratures can be reduced at the expense of increasing fluctuations in the conjugate quadrature. On the other hand, by entangling these two auxiliary modes, or instead employing a phase-sensitive amplifier, one could conceivably reduce the fluctuations within the two conjugate quadratures of the output field in unison.

Validity and approach:

The mathematical arguments presented in this work appear to be valid, logically sound, and there is sufficient information for the results to be reproduced. I found the writing of the manuscript easy to follow, and I commend the authors for their presentation of the results. However, there are some instances where I think this can be improved, and have noted these below.

Suggestions for improvement:

I think that some additional discussion of the quadrature operators $q = (a^\dagger + a)/\sqrt{2}$ and $p = i(a^\dagger - a)/\sqrt{2}$ should be given when they are introduced in Sec. I.C., to justify their usage as an approximation of the actual phase and amplitude quadratures of the field. Generally, this is not true as the actual steady state of the system (in the case of a phase-insensitive amplifier) has an undefined phase. For clarity of the reader, I believe that the authors should state their assumptions about the linearised treatment that they employ more explicitly here.

The discussion within the final paragraph of Sec. II.A. appears to contradict the results reported recently in the work: “No Tradeoff between Coherence and Sub-Poissonianity for Heisenberg-Limited Lasers, *Phys. Rev. Lett.* **130**, 183602 (2023)”, and its companion. There it was show that within a laser system, one could achieve maximum (Heisenberg-limited) coherence (reducing the phase noise), while at the same time reducing the fluctuations of the output power, by engineering the gain and output coupling.

I'm skeptical that the conclusions applying to "all feedback oscillators" are as general as claimed. I suspect this is nuanced, and could be fundamentally related to how one actually defines a "laser". I think discussion along these lines would improve clarity, from a theoretical point-of-view.

This notion of generality of this manuscript also relates to their Ref. [24]. In the final section of II.B. it is claimed that they extend the results of that paper to "all feedback oscillators". The paper they are referring to proved a number of results, and I think such a broad statement may be misleading to the reader. In reality, their system is not very general compared to what was proven in the paper they have cited. There, the ultimate limits to the properties of the output beam were derived from four axioms, which ensure all resources are correctly accounted for. Namely, the present systems only consist of a single mode that is being amplified, while the proof of the upper bound in Ref. [24] makes no such assumptions. I think some care is warranted in specifying what *exactly* from that paper is being improved upon; the statement "the results" seems too vague here.

Another recent work, that is highly relevant (but of which I could not find mention) is "Proposal for a continuous wave laser with linewidth well below the standard quantum limit. *Nat Commun* **12**, 5620 (2021)." There, the authors designed two physical systems capable of producing an output beam with a scaling beyond the ST limit ($\sim \mu^3$, to be precise). Since this is highly relevant to the present manuscript, I expected discussion of how it relates. Are there strengths of using a feedback oscillator with squeezed inputs and an amplifier, versus the approach reported there? Based on physical considerations, and accounting for all external sources of coherence, can a fair comparison be made?

I think more care should be taken to how the coherence is defined, since the definition is adopted from Ref. [24]. Here, it is taken to be simply the photon flux divided by the linewidth, rather than being the mean number of photons in the maximally populated mode of the output field. I do commend the authors for taking care in Appendix B to account for the additional coherent excitations that arise within the system, from the entangling the inputs.

In the abstract, the quantification of the efficacy of quantum resources in realising sub-SQL oscillators is mentioned. What are these quantum resources in the situation for the phase-sensitive amplifier discussed in Sec. II.C.? No discussion in the manuscript is really given about this point. The phase sensitive amplifier requires an additional phase reference, and this would require an external system that is adding coherent excitations to the system. An honest analysis of whether the system surpasses the SQL in this scenario would require one to account for these additional *coherent* excitations, as was done in the case of the entangled input in the preceding section (II.B.).

Significance:

Overall, I certainly think the results in the manuscript are interesting and worth publishing, but I am not convinced these results are noteworthy enough to warrant publication in Nature Communications. This is not to say that they don't provide some further understanding for the field—I certainly think they do, but they do seem rather incremental and perhaps not as general as claimed. I believe the manuscript, in its present form, would be much more suited to a more specialized journal. Also, given that this is

submitted to Nature Communications with theory results alone, I think there should be more emphasis on why their results are important. For example, what applications do the authors envisage these types of reduced-noise oscillators to have? Is there evidence of recent interest in these types of oscillators within the literature? Are these likely to motivate further experimental investigations of beams with post-SQL coherence, to a significant degree beyond that to which the literature discussed above will? I think these are reasonable standards one would expect a manuscript in Nat. Comms. to meet.

Some other minor comments:

- Statement above eq. (45) should read “... property (3) is satisfied”, not “... property (2) is satisfied”.
- Bottom of first column on pg. 10, first paragraph in C. should read: “ideal phase-insensitive amplifier”, not “ideal phase-sensitive amplifier”.
- I understand eq. (63) requires assumption of a vacuum input in auxiliary mode of beam splitter. In some parts of the paper, squeezed/entangled states are used as an input here. In the next paragraph this is mentioned, but it isn’t stated here that a vacuum input mode is being considered. Perhaps the authors could be more explicit and say that they are considering vacuum inputs to the two auxiliary gain and beam splitter modes.
- Above eq. (81), what is the old “saturation condition” explicitly? I understand this is $G = 1/\sqrt{\eta}$, but perhaps the equation number (46) should be referred to here.

Response to reviewer comments

We thank both reviewers for their thoughtful and constructive comments. In the following we give a detailed response (in blue) to their comments (in black). Many of our responses are also accompanied by changes to the enclosed manuscript, marked in blue in the enclosed manuscript.

Reviewer 1

The current version of manuscript is focused on the techniques, but the discussion of its real world connection is little. I suggest the authors to include more discussion on the practical side, including possible experimental setups that can verify the theoretical results, and explicit examples of potential application, e.g. engineering laser with lower phase noise, improving quantum information processors.

We modified the manuscript to address this concern in several ways. First, we modified the paper's introduction to point out the prevalence and importance of stable oscillators throughout many areas of physics, with a diverse array of references. Additionally, we have a new figure 1, which now includes several physical phase-insensitive and phase-sensitive feedback oscillators to make it clearer how our abstract model relates to the components of physical feedback oscillators of practical relevance.

I appreciate that the authors are deriving the results based on generic model and assumptions. However, while reading I could not stop making comparisons to the physical models of amplifiers I know. This could have subtle implications, as I explain later. Generally, I think it would be helpful to readers like me if the authors can include an example, at least in appendix, to show how the derivations agree with explicit physical models of amplifier (e.g. standard Hamiltonian and coupling of optical parametric oscillator).

We show by an explicit calculation (in Appendix D, "Comparison of an Ideal Purely Phase-Sensitive Oscillator to an Optical Parametric Oscillator"), that the output statistics of an OPO as derived from its Hamiltonian model and application of input-output theory, agrees with the results of the approach we follow in the main text (where the oscillator is modeled as a phase-sensitive amplifier in positive feedback).

At the beginning of p.2, the authors argue that the standard properties of linear amplifier "do not apply when the feedback loop is closed", they substantiate their argument by the fact that the gain will saturate. I do not fully agree with this argument. If one has a strictly linear amplifier, as the assumption of first paragraph of Sec. I, one can still construct such a feedback loop, except that the gain is exponential increasing and there is no steady state. The gain saturates is not due to the loop but the intrinsic non-linearity of the amplifier that becomes significant at large gain. One could also consider that the gain would saturate if the amplifier is sufficiently strongly driven, without introducing the loop.

We assume that a strictly linear amplifier is a mathematical idealization that has no counterpart in reality. For example, any amplifier (electronic or optical), has an energy source ("pump") that

drives the amplification mechanism; when the output goes larger than some value (related to the strength of the pump), the gain saturates (or reduces). This nonlinearity is indeed an intrinsic property of the amplifier. This nonlinearity is key in the study of oscillators, since, as the referee points out, this is what results in a finite steady state. What we meant to say in the paper was that this intrinsic nonlinearity becomes important when the amplifier is put in positive feedback, since in this case, there is a natural mechanism to drive the amplifier output towards saturation. We have clarified the language in the paper to address this confusion.

In Sec. IA, the authors present four generic assumptions of the amplifier. I find the assumption 4 might be too restrictive. I suppose it would be sufficient for the oscillator to reach a steady state if $dA/dx < 0$ for some sufficiently large x ? Assumption 4 seems to be suitable for laser-like system, which the photon output is determined by the fixed total number of atoms, but there could be other amplifiers (e.g. Josephson Parametric Amplifier) that the steady state is determined by the competition of multiple effects.

We addressed this comment by updating assumption 4 in the manuscript. As pointed out by the reviewer, the amplifier will reach a steady-state as long as we make the local assumption that $\frac{d^2A}{dx^2} < 0$ at the smallest value of x where $\frac{dA}{dx} = 1/\sqrt{\eta}$. Physically, this assumption is that at the lowest input value that the amplifier achieves a gain equal to the outcoupler's loss, the instantaneous gain, dA/dx , is a decreasing function of the input value. This assumption ensures that the amplifier reaches a steady-state that is stable against infinitesimal perturbations. While this local assumption ensures stability, the slightly stronger global assumption that $\frac{d^2A}{dx^2} < 0$ for all x seems to apply to most amplifiers and is simpler in its mathematical statement and physical interpretation; physically it requires that the amplifier's instantaneous gain decreases monotonically with the input level. This global assumption ensures stability against finite perturbations, not just infinitesimal perturbations as the local assumption does. We assume the global condition in the main text with a footnote explaining that the weaker assumption suffices for local stability.

In the phase insensitive model, I could understand that at the steady state the input and output are dominated by a strong displacement. By moving to the frame of the displacement, the leading order Hamiltonian in the amplifier will be quadratic in the mode operators, this gives the linear transformation in first line of Eq. 9. There are still higher-than-quadratic terms, as they must present for the existence of steady state, but their influence on quantum fluctuation is weak in most physical model of amplifier, thus the model in this manuscript is valid. However, the situation might not be the same in the phase sensitive case. It is known that squeezing can amplify the nonlinear interaction (see e.g. doi.org/10.1103/PhysRevLett.120.093601 and doi.org/10.1103/PhysRevLett.120.093602). If we consider the amplifier at strong phase sensitive amplification, in the squeezed frame the higher-than-quadratic terms are not necessarily negligible, so the input-output relation might not be linear. I believe the imperfect phase sensitive amplifier model in this manuscript is still valid if the phase insensitive gain dominates the overall gain, the authors should be more specific about the assumption and discuss its validity in physical systems. On the other hand, the "perfectly phase sensitive" case is doubtful to me. It is not clear if there could be self-consistent

model that the higher-than-quadratic terms can both contribute to the saturation of phase-sensitive gain and not inducing nonlinear transformation of the mode operators.

The premise of the PRLs cited by the referee is slightly different from our case. In the former, they analyse a parametric interaction of the form $\hbar g(a^\dagger b + ab^\dagger)$, where a is the ladder operator of an optical cavity mode, and b is the ladder operator of another system of interest (eg. atoms in cavity). If the cavity is further filled with a chi-squared medium, then the parametric interaction strength $\hbar g$ effectively becomes $\hbar g e^r$ (r the squeezing factor due to the chi-squared medium) – an exponential increase in coupling strength.

In our case, there is no appeal to a microscopic picture of an atomic medium in a cavity (further stuffed with a chi-squared medium). We presume the existence of a black-box amplifier which is linear and phase-sensitive for some region of the input amplitude. This regime is analysed using a Hamiltonian approach explicitly in the new Appendix D (“Comparison of an Ideal Purely Phase-Sensitive Oscillator to an Optical Parametric Oscillator”), and shown to agree with the results obtained using the approach elucidated in the main text.

Reviewer 2

I think that some additional discussion of the quadrature operators $q = (a^\dagger + a)/\sqrt{2}$ and $p = i(a^\dagger - a)/\sqrt{2}$ should be given when they are introduced in Sec. I.C., to justify their usage as an approximation of the actual phase and amplitude quadratures of the field. Generally, this is not true as the actual steady state of the system (in the case of a phase-insensitive amplifier) has an undefined phase. For clarity of the reader, I believe that the authors should state their assumptions about the linearised treatment that they employ more explicitly here.

We expanded upon figure 1 to include a “ball-and-stick” diagram showing how the quadrature operators geometrically approximate the phase and amplitude operators of the field when it has a large mean displacement from the origin. We also added a footnote which discusses technical issues in defining amplitude and phase operators and how the linearized treatment is well-justified in the large-field limit.

The discussion within the final paragraph of Sec. II.A. appears to contradict the results reported recently in the work: “No Tradeoff between Coherence and Sub-Poissonianity for Heisenberg-Limited Lasers, Phys. Rev. Lett. 130, 183602 (2023)”, and its companion. There it was shown that within a laser system, one could achieve maximum (Heisenberg-limited) coherence (reducing the phase noise), while at the same time reducing the fluctuations of the output power, by engineering the gain and output coupling.

This comment appears to misinterpret the results in that section of our paper. As we show later in the paper, it is possible to realize an oscillator with both intensity and phase fluctuations below the standard quantum limit for oscillator stability. What we show in Sec II.A is that phase-insensitive feedback oscillators with unentangled noise modes must obey an uncertainty product. However, by violating this assumption it is possible to evade the uncertainty product. In Sec. II.B and Sec. III, we relax both these assumptions and show that there need not be any tradeoff

between amplitude and phase noise. The paper quoted by the reviewer exploits yet another loophole, i.e. by using a highly nonlinear amplifier and out-coupler.

I'm skeptical that the conclusions applying to "all feedback oscillators" are as general as claimed. I suspect this is nuanced, and could be fundamentally related to how one actually defines a "laser". I think discussion along these lines would improve clarity, from a theoretical point-of-view.

If the motivation is to capture every aspect of a physical "laser" (whose nuances can change depending on what specific laser is considered) in a model, then it seems unlikely that such a model will apply to a sufficiently interesting class of oscillators.

Our motivation is not to study a "laser" per se, but take an approach that is as agnostic to the microscopic constitution of the oscillator as much as possible, while still capturing the essence of a feedback oscillator. (In this sense, we reflect the approach of Caves in his series of papers on amplifier noise.) Within this approach, we *define* feedback oscillators as systems where the output of a broadband amplifier is fed back via a delay loop. As illustrated in figure 1 (and Appendix D to a certain extent), this abstract definition captures the physics of oscillators of diverse composition and design.

This notion of generality of this manuscript also relates to their Ref. [24]. In the final section of II.B. it is claimed that they extend the results of that paper to "all feedback oscillators". The paper they are referring to proved a number of results, and I think such a broad statement may be misleading to the reader. In reality, their system is not very general compared to what was proven in the paper they have cited. There, the ultimate limits to the properties of the output beam were derived from four axioms, which ensure all resources are correctly accounted for. Namely, the present systems only consist of a single mode that is being amplified, while the proof of the upper bound in Ref. [24] makes no such assumptions. I think some care is warranted in specifying what exactly from that paper is being improved upon; the statement "the results" seems too vague here.

We rewrote the final paragraph of section II.B to more accurately compare our results to those of existing literature, which we cite in this paragraph. In particular, we note that our results agree with claims in recent literature that it is possible to realize oscillators with enhanced phase stability and amplitude stability beyond the Schawlow-Townes limit. However, existing literature shows that this is possible by engineering highly nonlinear oscillators. In contrast, our results show that this is also possible with linearized feedback oscillators by manipulating quantum states at various points in the loop.

This fact is relevant since strong nonlinearities cannot be as universally realized across all wavelengths as strongly squeezed fields can be (for example, UV and IR). Thus, we believe our results complement those of existing literature by providing an alternative scheme to realize oscillators with amplitude and phase stability beyond the Schawlow-Townes limit.

Another point of difference is that our statement of the standard quantum limit (SQL) for feedback oscillators, is given in terms of a product of the oscillator's output amplitude and phase

quadrature spectra, which are experimentally easy to obtain (say, by homodyne detection of the output fields). It is not so apparent how the notion of coherence introduced in recent work can be experimentally measured. In fact, coherence, as defined in ref. [24] is given by an integral over all time, or equivalently over all Fourier frequencies. This definition has several problems. First, it is impossible to measure across all time or all frequencies, so this notion of coherence cannot be determined from experiment. Secondly, while these integrals converge for the case of pure phase diffusion, which is assumed in ref. [24], they diverge for oscillators with a macroscopic output in a single field mode. Thus, the two approaches appear to complement each other depending on the scenario being considered.

Another recent work, that is highly relevant (but of which I could not find mention) is “Proposal for a continuous wave laser with linewidth well below the standard quantum limit. Nat Commun 12, 5620 (2021).” There, the authors designed two physical systems capable of producing an output beam with a scaling beyond the ST limit ($\sim \mu^3$, to be precise). Since this is highly relevant to the present manuscript, I expected discussion of how it relates. Are there strengths of using a feedback oscillator with squeezed inputs and an amplifier, versus the approach reported there? Based on physical considerations, and accounting for all external sources of coherence, can a fair comparison be made?

We thank the referee for pointing out this work, which we now cite in the final paragraph of section II.B. The article cited in this comment proposes a microwave oscillator operating below the SQL. It achieves this sub-SQL operation by engineering a strongly nonlinear amplifier and out-coupler. Our approach at achieving sub-SQL oscillator stability by manipulating the quantum states of the oscillator’s noise modes provides an alternative approach to engineer oscillators operating below the SQL. Accounting for external sources of coherence, both of these devices achieve μ^3 coherence scaling, as we discuss in Appendix B.

I think more care should be taken to how the coherence is defined, since the definition is adopted from Ref. [24]. Here, it is taken to be simply the photon flux divided by the linewidth, rather than being the mean number of photons in the maximally populated mode of the output field. I do commend the authors for taking care in Appendix B to account for the additional coherent excitations that arise within the system, from the entangling the inputs.

We moved the definition of coherence from the main text to an appendix and take it to be defined by the mean number of photons in the maximally populated mode of the output field as in ref. [24]. We then note that for the case of oscillators undergoing pure phase diffusion, this definition is equivalent to the definition in terms of linewidth, as discussed in ref. [24]. This pure phase diffusion case is relevant to the phase-insensitive feedback oscillators discussed in our paper, and using the definition in terms of linewidth simplifies calculations for these oscillators.

We also note that for the cases considered in our paper, the coherence is well defined for phase-insensitive feedback oscillators if we take the coherence to be given by the photon flux divided by the linewidth, but the integral in the definition in terms of the maximally populated mode does not exist. This issue appears to have similar roots to cases where it is possible to define an oscillator’s power spectrum, but the notion of linewidth (some weighted integral of the power spectrum) is either ambiguous or undefinable. Thus, we believe oscillator stability is more easily

defined in terms of the versatile notion of output frequency spectrum, and we use this to define the standard quantum limit.

The extension of the notion of coherence as in ref. [24] to scenarios other than phase diffusion seems to require further work, which is beyond the scope, and interest, of our paper.

In the abstract, the quantification of the efficacy of quantum resources in realising sub- SQL oscillators is mentioned. What are these quantum resources in the situation for the phase-sensitive amplifier discussed in Sec. II.C.? No discussion in the manuscript is really given about this point. The phase sensitive amplifier requires an additional phase reference, and this would require an external system that is adding coherent excitations to the system. An honest analysis of whether the system surpasses the SQL in this scenario would require one to account for these additional coherent excitations, as was done in the case of the entangled input in the preceding section (II.B.).

We address this comment briefly in the text in section II.C by noting that phase-sensitive feedback oscillators will inherently require an external phase reference in the form of a coherent pump field. Having said that, a comprehensive redressal of this important point would require a formulation and study of a “resource theory” of oscillators.

The referee’s comment may also be interpreted as suggesting “the SQL” should be taken to be the SQL as defined in “The Heisenberg Limit for Laser Coherence” (referred to as ref. [24] above). However, as pointed out above, this is not feasible since the single parameter of coherence does not seem to be well defined for some of the cases we consider.

So, it appears that a more flexible notion of coherence, and a resource theory built on that would fully answer the referee’s question. But clearly, this task does not detract from, or fall within the scope, of the current work.

Overall, I certainly think the results in the manuscript are interesting and worth publishing, but I am not convinced these results are noteworthy enough to warrant publication in Nature Communications. This is not to say that they don’t provide some further understanding for the field—I certainly think they do, but they do seem rather incremental and perhaps not as general as claimed. I believe the manuscript, in its present form, would be much more suited to a more specialized journal. Also, given that this is submitted to Nature Communications with theory results alone, I think there should be more emphasis on why their results are important. For example, what applications do the authors envisage these types of reduced-noise oscillators to have? Is there evidence of recent interest in these types of oscillators within the literature? Are these likely to motivate further experimental investigations of beams with post-SQL coherence, to a significant degree beyond that to which the literature discussed above will? I think these are reasonable standards one would expect a manuscript in Nat. Comms. to meet.

In the updated paragraph at the end of sec. II.B, we emphasize how our results yield a new pathway to post-SQL coherence. Existing literature emphasizes that post-SQL coherence can be achieved through strongly nonlinear amplifiers or out-couplers. Our results show it is also possible to achieve post-SQL coherence in *linearized* oscillators by manipulating their quantum

states. While existing literature focuses on implementing devices with post-SQL coherence in circuit QED where strong nonlinearities exist, we believe our approach will be highly relevant in other platforms where these nonlinearities are more challenging to achieve experimentally.

In the introduction, we now cite numerous examples of the importance in highly stable oscillators throughout modern physics and of ongoing interest in developing new methods of oscillator stabilization.

We also thank the referee for spotting a handful of typos, and minor typographic clarifications, in the manuscript, which we have addressed.

REVIEWERS' COMMENTS

Reviewer #1 (Remarks to the Author):

The authors have addressed all my concerns. I recommend it to be published as its current form.

Reviewer #2 (Remarks to the Author):

I can see that the author's have clearly taken care to consider the comments raised from both reviewers in the first round of review. With the exception of those I mention below, I am satisfied with the way most of these have been addressed. Below are the remaining comments and suggestions I have:

In their response, the authors added the paragraph including the sentence: *Recent work [51, 52] has shown that sub-Schawlow-Townes performance is also achievable with out-couplers that feature a nonlinear response to the fields that impinge it, and that such nonlinear systems need not trade off amplitude stability to realize enhanced phase stability.* This seems to me slightly confusing given the discussion in the previous section, where they say:

As long as the modes a_0 , a_G are independent, and the amplifier is phase-insensitive any attempt to reduce frequency fluctuations below the Schawlow-Townes limit — by engineering the out-coupler or ancillary states — will elicit increased fluctuations in the output power of the oscillator. Do the out-couplers require any caveats in the latter statement? Perhaps a small clarification can be made here to assist the reader.

Regarding the authors comment: *Another point of difference is that our statement of the standard quantum limit (SQL) for feedback oscillators, is given in terms of a product of the oscillator's output amplitude and phase quadrature spectra, which are experimentally easy to obtain (say, by homodyne detection of the output fields). It is not so apparent how the notion of coherence introduced in recent work can be experimentally measured. In fact, coherence, as defined in ref. [24] is given by an integral over all time, or equivalently over all Fourier frequencies. This definition has several problems. First, it is impossible to measure across all time or all frequencies, so this notion of coherence cannot be determined from experiment. Secondly, while these integrals converge for the case of pure phase diffusion, which is assumed in ref. [24], they diverge for oscillators with a macroscopic output in a single field mode. Thus, the two approaches appear to complement each other depending on the scenario being considered.* I certainly agree the two approaches complement each other. However, there are some points I take issue with here (these comments pertain to the authors response, not any content in the manuscript itself). The way coherence is defined from Ref. [24], is, in general, not the integral of $g^{(1)}$ over all time. It is the mean number of photons in the maximally populated mode of the output field, within a given frequency band. Regarding the statement of non-convergence of the integral, it seems to me that this comment stems from the fact that there is no such thing as a global phase reference, e.g. the source that is used to implement the phase-sensitive amplifier in their second model presented in the manuscript. In reality, global phase references don't exist, so such an integral would converge and could presumably be measured using heterodyne detection. Further, this assumption of a global phase reference, which the authors themselves use is the reason why they cannot compute that definition of coherence for their phase-sensitive feedback oscillator model.

The final issue is regarding the point I raised about the mention of the “quantification of the efficacy of quantum resources in realising sub- SQL oscillators”, as claimed in the abstract. The authors responded, *We address this comment briefly in the text in section II.C by noting that phase-sensitive feedback oscillators will inherently require an external phase reference in the form of a coherent pump field. Having said that, a comprehensive*

*redressal of this important point would require a formulation and study of a “resource theory” of oscillators. The referee’s comment may also be interpreted as suggesting “the SQL” should be taken to be the SQL as defined in “The Heisenberg Limit for Laser Coherence” (referred to as ref. [24] above). However, as pointed out above, this is not feasible since the single parameter of coherence does not seem to be well defined for some of the cases we consider. So, it appears that a more flexible notion of coherence, and a resource theory built on that would fully answer the referee’s question. But clearly, this task does not detract from, or fall within the scope, of the current work. I’m surprised to see that the authors responded to my comment by saying that accounting for this extra phase reference is outside the scope of their paper. First, I wasn’t suggesting that the authors adopt a different notion of coherence, or definition of sub-ST behaviour, but since they say in the abstract that they “[quantify] the efficacy of quantum resources in realizing sub-SQL oscillators”, I feel that this statement should be justified precisely, or at least emphasized, in section II.C. As the authors acknowledge, the resource which is giving rise to the reduction in phase fluctuations is coming from an external source, one which is assumed to have perfect phase stability. How is that quantified? Doing so would not entail developing an *entire* resource theory of oscillators. It seems that not much care is given to this aspect, but I feel it is very important, especially since I expected it from reading the abstract.*

Response to referee #2

1. In their response, the authors added the paragraph including the sentence: *Recent work [51, 52] has shown that sub-Schawlow-Townes performance is also achievable with out-couplers that feature a nonlinear response to the fields that impinge it, and that such nonlinear systems need not trade off amplitude stability to realize enhanced phase stability.* This seems to me slightly confusing given the discussion in the previous section, where they say:

As long as the modes a_0 , a_G are independent, and the amplifier is phase-insensitive any attempt to reduce frequency fluctuations below the Schawlow-Townes limit — by engineering the out-coupler or ancillary states — will elicit increased fluctuations in the output power of the oscillator. Do the out-couplers require any caveats in the latter statement? Perhaps a small clarification can be made here to assist the reader.

We now explicitly mention the linear nature of the output coupler in this context.

2. Regarding the authors comment: *Another point of difference is that our statement of the standard quantum limit (SQL) for feedback oscillators, is given in terms of a product of the oscillator's output amplitude and phase quadrature spectra, which are experimentally easy to obtain (say, by homodyne detection of the output fields). It is not so apparent how the notion of coherence introduced in recent work can be experimentally measured. In fact, coherence, as defined in ref. [24] is given by an integral over all time, or equivalently over all Fourier frequencies. This definition has several problems. First, it is impossible to measure across all time or all frequencies, so this notion of coherence cannot be determined from experiment. Secondly, while these integrals converge for the case of pure phase diffusion, which is assumed in ref. [24], they diverge for oscillators with a macroscopic output in a single field mode. Thus, the two approaches appear to complement each other depending on the scenario being considered.* I certainly agree the two approaches complement each other. However, there are some points I take issue with here (these comments pertain to the authors response, not any content in the manuscript itself). The way coherence is defined from Ref. [24], is, in general, not the integral of $g^{(1)}$ over all time. It is the mean number of photons in the maximally populated mode of the output field, within a given frequency band. Regarding the statement of non-convergence of the integral, it seems to me that this comment stems from the fact that there is no such thing as a global phase reference, e.g. the source that is used to implement the phase-sensitive amplifier in their second model presented in the manuscript. In reality, global phase references don't exist, so such an integral would converge and could presumably be measured using heterodyne detection. Further, this assumption of a global phase reference, which the authors themselves use is the reason why they cannot compute that definition of coherence for their phase-sensitive feedback oscillator model.

In order to compute the “mean number of photons in the maximally populated mode of the output field, within a frequency band” in an unambiguous fashion, the power spectrum has to be integrated in some frequency band, with an arbitrary cutoff delineating one mode from its neighbour. It seems to us that this arbitrariness is no better than the arbitrariness inherent in the choice of a phase reference. The one way to remove this ambiguity, by integrating over all frequencies, leads to the convergence issues we raised.

3. The final issue is regarding the point I raised about the mention of the “quantification of the efficacy of quantum resources in realising sub- SQL oscillators”, as claimed in the abstract. The

authors responded, *We address this comment briefly in the text in section II.C by noting that phase-sensitive feedback oscillators will inherently require an external phase reference in the form of a coherent pump field. Having said that, a comprehensive redressal of this important point would require a formulation and study of a “resource theory” of oscillators. The referee’s comment may also be interpreted as suggesting “the SQL” should be taken to be the SQL as defined in “The Heisenberg Limit for Laser Coherence” (referred to as ref. [24] above). However, as pointed out above, this is not feasible since the single parameter of coherence does not seem to be well defined for some of the cases we consider. So, it appears that a more flexible notion of coherence, and a resource theory built on that would fully answer the referee’s question. But clearly, this task does not detract from, or fall within the scope, of the current work.* I’m surprised to see that the authors responded to my comment by saying that accounting for this extra phase reference is outside the scope of their paper. First, I wasn’t suggesting that the authors adopt a different notion of coherence, or definition of sub-ST behaviour, but since they say in the abstract that they “[quantify] the efficacy of quantum resources in realizing sub-SQL oscillators”, I feel that this statement should be justified precisely, or at least emphasized, in section II.C. As the authors acknowledge, the resource which is giving rise to the reduction in phase fluctuations is coming from an external source, one which is assumed to have perfect phase stability. How is that quantified? Doing so would not entail developing an entire resource theory of oscillators. It seems that not much care is given to this aspect, but I feel it is very important, especially since I expected it from reading the abstract.

The word “resource” has acquired a specific, and precise, meaning as used in the context of “resource theories”. In order to avoid confusion, we now use the phrase “quantum strategies” in lieu of “quantum resources”.